# Self-supervised context-aware Covid-19 document exploration through atlas grounding

**Dusan Grujicic**[*] **Gorjan Radevski**[*], **Tinne Tuytelaars, and Matthew B. Blaschko**
ESAT-PSI, KU Leuven, Kasteelpark Arenberg 10, 3001 Leuven, Belgium
{firstname.lastname}@esat.kuleuven.be

## Abstract

In this paper, we aim to develop a self-supervised grounding of Covid-related medical text based on the actual spatial relationships between the referred anatomical concepts. More specifically, we learn to project sentences into a physical space defined by a three-dimensional anatomical atlas, allowing for a visual approach to navigating Covid-related literature. We design a straightforward and empirically effective training objective to reduce the curated data dependency issue. We use BERT as the main building block of our model and perform a comparison of two BERT variants pre-trained on general-purpose text - BERT$_{BASE}$ and BERT$_{SMALL}$, with three domain-specific pre-trained alternatives - BIOBERT, SCIBERT and CLINICALBERT. We perform a quantitative analysis that demonstrates that the model learns a context-aware mapping while being trained with self-supervision in the form of medical term occurrences. We illustrate two potential use-cases for our approach, one in interactive, 3D data exploration, and the other in document retrieval. To accelerate research in this direction, we make public all trained models, the data we use, and our codebase. Finally, we also release a web tool for document retrieval and a visualization tool.

## 1 Introduction

The quantity of available COVID-19 articles on the internet increases every day. Nonetheless, it remains scarce compared to general domain data sets. Annotating medical data requires the expertise of physicians, and is therefore cost-prohibitive, especially during a pandemic. As a consequence of the lack of available structured data in the medical domain, the machine learning community has mostly focused on developing general-purpose text models.

---

[*]equal contribution

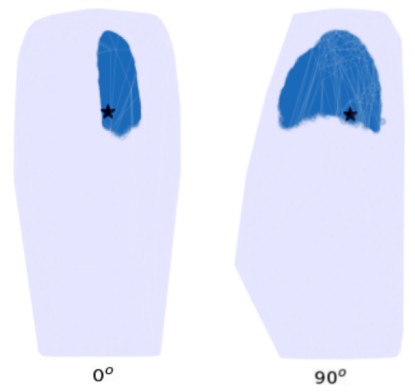

Figure 1: Grounding of the sentence "The total number of AM in the lung at 48 hr was significantly ($p < 0.05$) reduced as compared to PBS controls" (Hartwig et al., 2014) with the CLINICALBERT model. The dark blue represents the voxels of the left lung, while the light blue area represents the outline of the body. The star denotes the model prediction. See Section 6 for details.

The development of BERT (Devlin et al., 2018), and the increased popularity of transfer learning in natural language processing (NLP), have prompted notable works that aim to leverage publicly available medical and scientific articles to develop domain specific pre-trained language models (Lee et al., 2019; Alsentzer et al., 2019; Beltagy et al., 2019). These approaches train models that learn universal sentence embeddings aimed at capturing the semantics and structure of the text data.

In contrast, we focus on mapping text to locations in a 3D model of the human body (Figure 1), where the physical proximity of objects reflects their functional and semantic relatedness to a significant degree. Such an embedding is advantageous for several reasons: (i) It allows us to visualize medical text in physically meaningful space, finding clusters of documents organized by anatomy (Figure 5). (ii) It allows us to search for and re-

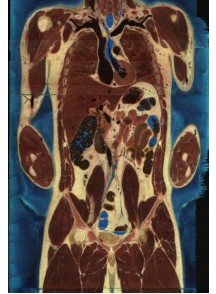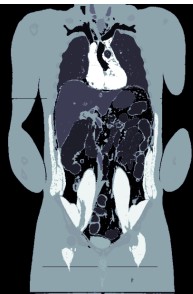

Figure 2: Cross-sections of the RGB volume (left) and the grayscale volume representing segmentation labels (right) (Pommert et al., 2001)

trieve text by navigating through a physical space. (iii) There is a statistical advantage to modelling medical text in the 3D space as anatomically related substructures tend to be close to one another.

In the absence of semantic labels, we use term occurrences as the indication of what the text denotes. For example, the sentence: "The **pancreas** contains tissue with an endocrine and exocrine role" receives a target of mapping to the location of the pancreas in the 3D space.

In order to achieve the goal of grounding medical text into the physical space, a reference location for every medical term of interest is required. Such references can be obtained from a combination of a three-dimensional atlas of human anatomy and contextual information. There are multiple digital anatomical models available. The Virtual Population (Christ et al., 2009; Gosselin et al., 2014) of the *IT'IS Foundation*[1] contains anatomical models of 10 different persons obtained from MRI procedures. The **S**egmented **I**nner **O**rgans **(SIO)** from the *Voxel-Man* project (Höhne et al., 2001; Pommert et al., 2001; Schiemann et al., 1997) [2] is based on the *Visible Human Male* (U.S. National Library of Medicine [3]) and contains 202 labeled anatomical objects within the human torso. The model consists of 774 slices obtained by CT and MRI imaging, where each slice contains a cryosection image, a CT image and a segmentation label image where the grayscale level corresponds to a segmentation label of the tissue (Figure 2). In this work, we build on the atlas of Pommert et al. (2001), though the approach is readily extended to other atlases.

[1] www.itis.swiss/
[2] www.voxel-man.com/
[3] www.nlm.nih.gov/research/visible/

## 2 Related Work

There are many works that deal with sentence grounding in a limited space, albeit not in the physical 3D space as we do. Most of the approaches exploit multimodal data and limit the projection space to either images or videos (Akbari et al., 2019; Kiela et al., 2017; Chen et al., 2019; Javed et al., 2018; Xiao et al., 2017). These works overcome expensive bounding box or pixel-level annotation, but they cannot be extended to the unsupervised setting where the data are not paired, but rather raw unpaired sentences or images. Even though the image-caption pairs without any region label are commonly referred to as weakly-supervised data in the literature, most of these works have training procedures that are dependent on curated datasets which are hard to obtain.

The works of Weyand et al. (2016); Ebrahimi et al. (2018) are probably the most similar to ours. In PlaNet, Weyand et al. (2016) attempt to classify images to a distinct set of geolocations. To do so, they train their model on a dataset of geotagged images where each image belongs to a single class: the index of the partitioned geolocation world cell. In contrast to our approach, the task is formulated as a classification problem where the physical distances and relationships between cells do not affect the way the probability distribution over them is learned. We frame our approach as a regression problem, as the spatial closeness of anatomical concepts implies a degree of semantic and functional affinity. This helps us reason about our approach in a way that in addition to knowing whether the grounding is correct or not, we have insight into how physically close we are to the target.

A similar approach, but more related to our work as it also deals with text, is the work of Ebrahimi et al. (2018), where the extracted text representations and metadata were used to classify tweets by geographic region in a fully-supervised setting. Ebrahimi et al. (2018) utilize machine learning to ground sentences in the world atlas. Yet and again, their approach is dependent on a carefully structured dataset and availability of explicit annotations. In our work, we attempt to go one level further and learn to ground sentences by only using unlabeled raw text data, obtained from medical journal articles, while preserving the spatial structure of the sentences. Our supervision comes in the form of implicit organ voxel points in a human atlas space, and words/phrases that make reference to those or-

gans. To the best of our knowledge, so far, there have not been works that attempt to ground sentences in a 3D human atlas space, using strictly self-supervision. Additionally, a number of works have applied natural language processing techniques on Covid-19 articles (Zhang et al., 2020; Wang et al., 2020; Liang and Xie, 2020), however, none of them aim to ground text in the 3D atlas space.

## 3 Methods

In this section, we describe the model we use which is entirely based on BERT, the training objective and the task that we address in this paper.

### 3.1 The model

**B**idirectional **E**ncoder **R**epresentations from **T**ransformers - BERT (Devlin et al., 2018) is a pre-trained Transformer (Vaswani et al., 2017) based language model. Before BERT, the only way to train deep bidirectional language models was to train a separate forward and backward language model, and in the end, concatenate their learned representations (Peters et al., 2018). BERT alleviates that problem by introducing the concept of Masked Language Modelling (MLM), previously known as *cloze* task (Taylor, 1953). The scalability of BERT, combined with MLM, led to the increasing popularity of such language models (Keskar et al., 2019; Liu et al., 2019; Lample and Conneau, 2019).

Due to the train-test discrepancy that occurs by including the **[MASK]** token in the MLM, other approaches train transformers in an autoregressive manner (Radford et al., 2019; Yang et al., 2019; Dai et al., 2019). In our work, we use BERT as a backbone in our model due to its simplicity and applicability in a wide range of domains. As we shall see later when we describe the task of text-atlas grounding, the existence of the **[MASK]** token in the vocabulary can be seamlessly incorporated in our pipeline to fit within the task we solve. In our work, we perform an ablation study with five different pre-trained BERT models. Following the standard practice (Devlin et al., 2018), we take the representation of the **[CLS]** token as a general representation of the whole sequence. Finally, to obtain the 3D atlas grounding for a piece of medical text, we project BERT'S sentence embedding with a linear layer, mapping from BERT'S hidden space to the 3D space.

### 3.2 Text-to-atlas mapping objective

Our final objective is to ground medical texts to the anatomic atlas space using only self-supervision in the form of organ appearances in each sentence. More concretely, we have a dataset of sentences, where for each sentence, we can detect the appearances of terms denoted in the human atlas. Then, our desired scenario is that sentences that share the same semantics are mapped in the same region in the human atlas space regardless of whether they make explicit reference to an organ. To achieve that, we tokenize each of the training sentences (Loper and Bird, 2002) and stochastically mask each of the keywords. Each of the keywords (organs) is masked with 0.5 probability. In other words, assuming that we have the sentence "In addition, the kidney mRNA transcript level and serum activity of XOD in the infected group was significantly higher than that of the control group at 8, 15 and 22 dpi ($p < 0.05$)" (Lin et al., 2015) on average, 50% of the time we will replace it with "In addition, the **[MASK]** mRNA transcript level and serum activity of XOD in the infected group was significantly higher than that of the control group at 8, 15 and 22 dpi ($p < 0.05$)" in the current training batch. We use the **[MASK]** token, as it is included in BERT'S default vocabulary. Next, the sentence words are joined again and tokenized using the WordPiece (Wu et al., 2016) tokenization method as per Devlin et al. (2018). By following the above-mentioned procedure, we are able to obtain context-dependent grounding, such that the model can ground sentences purely based on their context in cases where none of the organ references are present.

### 3.3 Minimum organ distance loss

Ideally, if we had exactly one organ occurrence per sentence, and if we could associate each organ with a single point in the 3D space, we could simply minimize the mean squared error between the 3D coordinates of the organ point $y$ and the model prediction $\hat{y}$. However, a sentence can contain multiple organ occurrences, while organs themselves are distributed in nature, and are characterized by a set of points in 3D space, which capture its position, size and shape. Therefore, the loss function needs to accommodate having more than one point as target for regression.

We calculate the Euclidean distances between the prediction and each organ point, and the soft-min (soft-max across the inputs reversed in sign)

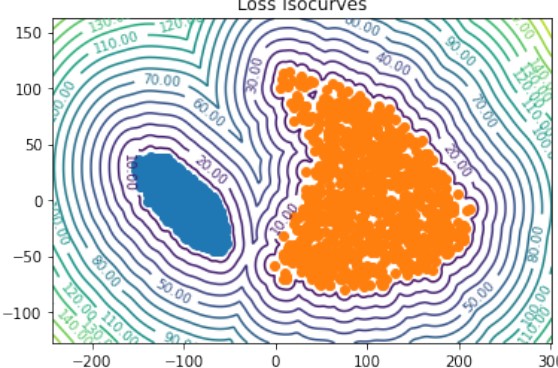

Figure 3: Loss isocurves around kidney and lung point clouds projected into 2D using PCA for visualization purposes.

across these squared distances as weights for the contributions of individual points. The loss contribution of an organ point (denoted as $PC$) is the product of its squared distance from the predicted point and its weight:

$$PC(y_p) = \frac{\|y_p - \hat{y}\|_2^2 \exp(-\gamma_1\|y_p - \hat{y}\|_2^2)}{\sum_{i=1}^{P} \exp(-\gamma_1\|y_i - \hat{y}\|_2^2)}, \quad (1)$$

where $\hat{y}$ is the model prediction, $y_p$ is an organ point, $P$ is the total number of points that characterize a single organ and $\gamma_1$ is a temperature term. We calculate the loss for one organ (denoted as $OL$) as the sum of contributions of its points:

$$OL = \sum_{p=1}^{P} PC(y_p) \quad (2)$$

To avoid regressing to a point outside of the organ, we shave off the surface of the organ by performing a single binary morphological erosion (Serra, 1983) prior to computing the loss.

In the case where more than one organ is present in the sentence, we calculate the loss for each individual organ in the way described above. Then, we compute the soft-min over the set of such loss terms as contribution weights for each organ. The final loss contribution of one organ (denoted as $OC$) is the product between its individual loss and its contribution weight:

$$OC_o = \frac{OL_o \exp(-\gamma_2 OL_o)}{\sum_{i=1}^{O} \exp(-\gamma_2 OL_i)} \quad (3)$$

where $O$ is the total number of distinct organs appearing in the sentence, $OL_i$ is the organ loss for the $i$th organ, and $\gamma_2$ is a temperature term. Finally, the total loss for one sample is computed by summing up the loss contributions of organs appearing in its sentence:

$$Loss = \sum_{o=1}^{O} OC_o \quad (4)$$

## 4 Data Collection

### 4.1 Text Corpus

The text corpus consists of Covid-19 related articles from the Open Research Dataset Challenge (**CORD-19**) [4]. The version from 20.03.2020., consisting of a csv file with metadata for 29500 papers and 13202 json files with full texts of scientific articles pertaining to Covid-19 was used for training the model. The abstracts and text bodies of full text articles were included in the corpus and split into sentences, which constitute the samples in the dataset. Both the full text json files and the metadata csv contain paper abstracts, and in case when there is a mismatch between the two, we include the abstract version that contains more characters. The sentence length was analyzed and it was found that $99.89\%$ sentences contain fewer than 128 words. In order to avoid unnecessary memory consumption during training, sentences longer than 128 words were discarded.

### 4.2 Human Body Atlas

We utilize the **S**egmented **I**nner **O**rgans (**SIO**) atlas (Pommert et al., 2001). We base the 3D atlas on the segmentation labels of the tissues in the human body, which come in the form of image slices that form a 3D voxel model of the male torso when stacked on top of one another. The stacked images from the torso represent a volume of $573 \times 330 \times 774$ voxels, with 1-millimeter resolution along each axis. The value of each voxel represents the segmentation label of its corresponding organ or tissue. The **SIO** includes the model of the human head as well, that we do not use.

SIO contains a glossary of medical terms and their associated segmentation labels. A list of synonyms and closely related wordforms for each glossary term were retrieved. The ScispaCy UmlsEntityLinker (Neumann et al., 2019) was used for searching the UMLS Metathesaurus *(The Unified Medical Language System)* (Bodenreider, 2004) for

---

[4] https://www.kaggle.com/allen-institute-for-ai/CORD-19-research-challenge

all word forms of the SIO glossary [5]. The parameters of the UmlsEntityLinker were kept at default values.

SIO includes 202 anatomical objects with their distinct segmentation labels. Tissues such as skin, gray matter, white matter, and unclassified tissues were removed from the set of labeled terms, as they denote general medical concepts not characterized by specific compact locations in the human body. The vertebrae, bones, and muscles of the arms and legs were discarded as well. In the case of categories for bilateral organs located symmetrically on both the left and the right side of the body, which are seldom mentioned explicitly in the texts, only the atlas voxels pertaining to the left organ were kept for every bilateral pair. Atlas labels that appear infrequently in medical literature, but are functionally related to other, more frequently occurring organs, or are colloquially referred to under a single, umbrella term, were merged. The aforementioned steps reduced the list of distinct anatomical objects of interest to 67. The full list of organ removals, mergers and renamings can be found at `https://github.com/gorjanradevski/macchina/`.

### 4.3 Dataset Creation

Sentences were chosen as the main units of text that are mapped to three-dimensional locations in the atlas, i.e. the samples consist of sentences and their targets in the human atlas. The voxels of one organ can be characterized by a point cloud in the atlas space, where each point represents the coordinate indices of one voxel (Figure 1).

The training set consists of sentences from 70% randomly chosen documents, while the remaining 30% of the documents were evenly distributed between the validation and the test set. Consequently, the sentences from the same document are always assigned to the same dataset split. As can be seen on Figure 4, the frequency of the words and phrases referring to the lung, liver, bronchi, stomach, and kidney is significantly higher than that of other organs. Therefore, to balance out the numbers of organ occurrences in the dataset, we include up to 8000 randomly selected sentences that contain these frequently occurring organs and discard the rest, while keeping all the sentences containing less frequently occurring organs. Some sentences contain multiple occurrences of one or different organs, meaning that an organ can still have more

---

[5]ScispaCy version 0.2.3 and en_core_sci_lg pipeline

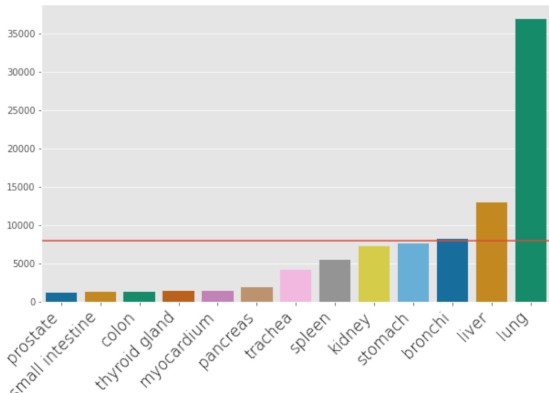

Figure 4: Number of occurrences of the 13 most frequent organs

than 8000 occurrences in the dataset. Regardless of this, the number of sentences that contain the most frequently occurring organs is significantly reduced, whereas the sentences containing less frequently occurring organs are preserved. The organs with fewer than 100 occurrences are removed. This included 38 organs, leaving a total of 29 anatomical categories as target locations for text mapping. The sentences that do not contain words and phrases that can be associated with the SIO glossary terms are discarded.

## 5 Experimental setup

For the development of our models and pipelines we used the PyTorch library (Paszke et al., 2019) together with the HuggingFace transformers (Wolf et al., 2019) package. For each of the experiments, we start with a pre-trained model and we fine-tune the whole architecture. We keep a fixed learning rate of $2 \times 10^{-5}$ and train the larger models for 20 epochs, and we increase the learning rate to $5 \times 10^{-5}$ for the BERT$_{\text{SMALL}}$ model and train it for 50 epochs. During training we clip the gradients when the global norm exceeds 2.0. For all experiments, our optimizer of choice is Adam (Kingma and Ba, 2014) and the temperature term $\gamma_1$ is fixed to 0.33. We fix the second temperature term $\gamma_2$ to $\frac{1}{N}$ where $N$ is the number of distinct organs appearing within an single training instance. During the fine-tuning we keep the model that reported the lowest distance to the nearest ground truth voxel point on the validation set as early stopping. Aside from early stopping and the dropout (Srivastava et al., 2014) layers present in BERT, we do not perform any other regularization.

## 5.1 Metrics and evaluation

We perform all evaluations in two different settings, namely Regular and Masked. In the former, we perform atlas grounding on a holdout set of sentences obtained from documents not seen by the model during training. In the latter, we use the same model while masking all the SIO glossary terms and their synonyms, i.e. substituting them with the special token **[MASK]**. In the Masked setting, we ensure that the model relies on the sentence context instead of making a one-to-one correspondence between the organ that appears in the sentence and the location in the atlas.

Each of the models is evaluated on three metrics: (i) Distance to the nearest voxel of the nearest correct organ (NVD)[6]. (ii) Distance to the nearest correct organ voxel calculated only on the samples for which the projection is outside the organ volume (NVD-O)[6]. (iii) Rate at which the sentences are grounded within the volume of the correct organ, which we denote as Inside Organ Ratio (IOR).

We consider the predicted 3D point to be inside the organ volume (*hit*) when its coordinates, rounded to the nearest integer to represent voxel indices, are within the set of voxels that make up the corresponding organ. In cases where the sentence has more than one organ reference, due to the implicit labeling, we measure a *hit* when the predicted coordinates correspond to any one of the given organs.

When the projection is inside the volume of the organ, the NVD is zero, and otherwise, it is measured as the distance to the surface of the nearest organ in the sentence. The NVD-O metric complements the NVD metric, such that it gives insight into how far off the prediction is when it misses the correct organ.

We justify the evaluation metrics according to the type of data we use, and the use-cases. Firstly, since we leverage unlabeled data exclusively, we assume that a single sentence needs to be grounded inside/near the organ of reference in the sentence. Secondly, we want similar sentences (sentences making reference to a certain body part), to be grounded in similar parts of the human atlas. As a result, we use the distance to the nearest organ voxel as the primary evaluation metric. Therefore, we can expect that the models with high evaluation scores to be useful for data-exploration and document retrieval through the human atlas.

---

[6]Calculated in centimeters

## 6 Quantitative results

In this section we report the results of our trained models. Four of the models share the same architecture, with the only difference being the pre-training corpus of BERT. Namely, the BERT$_{BASE}$ (Devlin et al., 2018) model has been pre-trained on the BooksCorpus (Zhu et al., 2015) and English Wikipedia. The BIOBERT (Lee et al., 2019) model is obtained by fine-tuning BERT$_{BASE}$ on PubMed abstracts and PMC full-text articles as per Lee et al. (2019) while CLINICALBERT is obtained by initializing with the BIOBERT'S weights and fine-tuning on clinical notes. The SCIBERT model is obtained by fine-tuning BERT$_{BASE}$ on 1.14M papers from Semantic Scholar (Ammar et al., 2018). Finally, BERT$_{SMALL}$, is obtained by pre-trained distillation (Turc et al., 2019) from BERT$_{BASE}$.

Additionally, we perform an analysis of the effectiveness of framing the task as classification. Here, we feed the **[CLS]** representation to an output layer to perform the classification as per Devlin et al. (2018). The model is trained to predict an organ index for every sentence, and the center of the predicted organ is subsequently used as the model prediction and evaluated in the same way as the regression models. We denote this model as CLASSCENTER in the result tables.

Finally, we report the results on two naive baselines that aim to exploit the information on the general locations of the organs and the information on the disbalance in the frequency of organ occurrences that exist in the datasets. In the first baseline (FREQUENCY), we measure the frequency of the organ terms in the training set samples, and always predict the point within the most frequent organ on the test set samples. In the second baseline (CENTER), we use the center of the 3D atlas as the prediction and measure the distance to the closest correct organ for every test sample (the IOR is not relevant).

## 7 Use-cases

By grounding medical sentences in a 3D atlas space, we produce low dimensional sentence embeddings. We discuss two use-cases of our model, which, either implicitly or explicitly, leverage such embeddings: (i) atlas based point-cloud corpus visualization and (ii) atlas based document retrieval.

We built a tool for each of the use cases, one for visualizing and retrieving articles in the text corpus by specifying 3D coordinates, and one for

| Method | Regular | Masked |
|---|---|---|
| BERT | $0.33 \pm 0.02$ | $3.31 \pm 0.08$ |
| BIOBERT | $0.21 \pm 0.02$ | $2.92 \pm 0.08$ |
| SCIBERT | $0.22 \pm 0.02$ | $3.33 \pm 0.09$ |
| BERT$_{SMALL}$ | $0.51 \pm 0.03$ | $3.44 \pm 0.08$ |
| CLINICALBERT | $0.25 \pm 0.02$ | $3.11 \pm 0.08$ |
| CLASSCENTER | $0.03 \pm 0.01$ | $1.66 \pm 0.07$ |
| CENTER | $10.77 \pm 0.10$ | $10.77 \pm 0.10$ |
| FREQUENCY | $9.49 \pm 0.15$ | $9.49 \pm 0.15$ |

Table 1: **NVD on the Cord-19 dataset -** we can infer that all models where the backbone is BERT$_{BASE}$ perform comparable to each other. BERT$_{SMALL}$ performs worse compared to the other models, and the smaller capacity makes the model unable to sufficiently fit the data. The CLASSCENTER model outperforms the rest of the models since it solves an easier task i.e. predicting a discrete value corresponding to the organ.

| Method | Regular | Masked |
|---|---|---|
| BERT | $4.6 \pm 0.26$ | $7.26 \pm 0.16$ |
| BIOBERT | $0.99 \pm 0.08$ | $5.99 \pm 0.15$ |
| SCIBERT | $2.27 \pm 0.18$ | $7.7 \pm 0.17$ |
| BERT$_{SMALL}$ | $2.11 \pm 0.1$ | $6.05 \pm 0.14$ |
| CLINICALBERT | $2.69 \pm 0.21$ | $7.5 \pm 0.18$ |
| CLASSCENTER | $24.94 \pm 6.26$ | $12.75 \pm 0.34$ |
| CENTER | $10.77 \pm 0.10$ | $10.77 \pm 0.10$ |
| FREQUENCY | $11.63 \pm 0.17$ | $11.63 \pm 0.17$ |

Table 2: **NVD-O on the Cord-19 dataset -** compared to NVD, here we can observe the main shortcoming of the CLASSCENTER model. Namely, when the model fails to predict the correct organ, the error is not mitigated by predicting a point in the vicinity of the correct organ, as is the case with models that ground sentences by projecting them to the 3D atlas.

retrieving relevant articles based on a textual query. The data was obtained from the Covid-19 Open Research Dataset Challenge (CORD-19) [7] hosted on Kaggle. The version from 10.04.2020., consisting of 59311 json files of scientific articles pertaining to Covid-19 and metadata for 51078 papers was the latest at the time of writing. The dataset was processed by using paper indexes for matching titles, abstracts and main texts in the json files with the information required for retrieving the article in the metadata. This included the source of the publication, authors, date, digital object identifier (DOI) and the URL for each paper - all the relevant information for article retrieval. In the case of both tools, each document abstract was embedded into the 3D space as a point cloud, where each point is the output of the model for each of its sentences. Tools and code can be accessed at https://github.com/gorjanradevski/macchina/.

### 7.1 Atlas based point-cloud corpus visualization

One advantage of text retrieval in the physical 3D space is that we do not need to use textual queries, but are also able to retrieve information by directly specifying an observable desired location in the human atlas space. Another advantage is being able to directly observe the relationship between

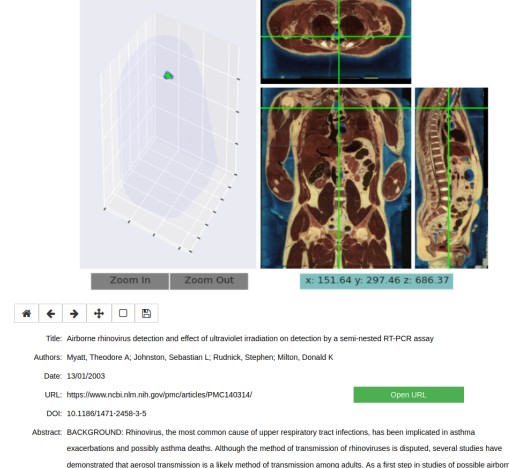

Figure 5: Point-cloud corpus visualization tool.

embedded texts in an intuitively meaningful setting.

The point based tool (Figure 5) accepts a query in the form of 3D coordinates and matches articles based on the proximity of their embeddings in 3D space. The 3D point is queried by selecting a 2D point on two out of three orthogonal cross-sections. The distance between the queried point and the embedded articles is calculated as the distance between the query point and the centroids of article point clouds. The nearest 50 articles are shown as the centroids of their sentence point clouds in the 3D view on the left, allowing the user to navigate between the closest suggestions. The user may zoom in and click on nearby points, after which the information on the corresponding article is displayed.

---

[7]https://www.kaggle.com/allen-institute-for-ai/CORD-19-research-challenge/

| Method | Regular | Masked |
|---|---|---|
| BERT | $92.76 \pm 0.29$ | $54.44 \pm 0.56$ |
| BIOBERT | $78.21 \pm 0.47$ | $51.31 \pm 0.57$ |
| SCIBERT | $90.24 \pm 0.34$ | $56.76 \pm 0.56$ |
| BERT$_{SMALL}$ | $75.75 \pm 0.48$ | $43.21 \pm 0.56$ |
| CLINICALBERT | $90.89 \pm 0.33$ | $58.56 \pm 0.56$ |
| CLASSCENTER | $99.88 \pm 0.04$ | $86.96 \pm 0.38$ |
| CENTER | $0.00 \pm 0.00$ | $0.00 \pm 0.00$ |
| FREQUENCY | $18.41 \pm 0.44$ | $18.41 \pm 0.44$ |

Table 3: **IOR on the Cord-19 dataset** - When evaluated on the Inside Organ Ratio, the CLASSCENTER model, since it directly optimizes the IOR metric, significantly outperforms all others. Even though the grounding models approximate this metric during the training process, we can observe that for most of the models, the IOR exceeds 90% in the Regular setting and 50% in the Masked setting.

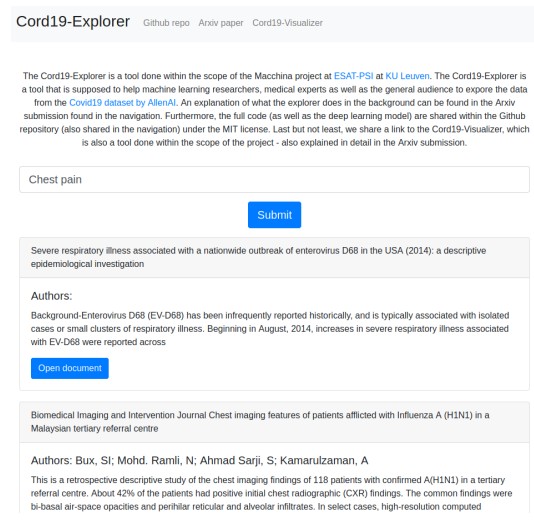

Figure 6: Text based document retrieval tool.

## 7.2 Atlas based document retrieval

The text query based tool (Figure 6) accepts a text query, tokenizes it into sentences and embeds each into a point in the 3D space, creating a point cloud. The embedded point cloud is compared with the point clouds of embedded abstract sentences of each article. The articles are ranked in terms of the distances between the point cloud centroids. The information on the 200 closest articles is retrieved, and it consists of the title, abstract and the link to the publication.

## 8 Discussion and Conclusions

There are several shortcomings in the current study. First, we only utilized a single male atlas to compute embeddings. Future work should explore multiple embeddings based on different age, gender, and body type (Christ et al., 2009; Gosselin et al., 2014). Additionally, the choice of labels for the atlas was determined separately from the specific task of Covid-19 article embeddings, and may have suboptimal levels of granularity in labeling organ substructures for this specific task. Second, for expedience we only explored training on individual sentences, as opposed to larger bodies of text with label propagation from nearby sentences. Third, we have formulated sentence embeddings in an atlas as a prediction of a single point, but we could also have considered predicting a (multi-modal) distribution over the atlas space per sentence. Finally, the query tools would ideally be validated with a user study. In the current crisis, the medical experts who would form the user group are in high demand, and we therefore postpone this step pending their availability.

In this paper, we have presented a self-supervised approach to ground medical texts in a 3D human atlas space. We have relaxed the labeled data constraint and provided an objective that learns semantically aware groundings of sentences. We did an ablation study of the performance on the sentence grounding task with 5 different BERT backbone models, namely the standard BERT as per Devlin et al. (2018), BIOBERT (Lee et al., 2019), SCIBERT (Beltagy et al., 2019), CLINICALBERT (Alsentzer et al., 2019) and BERT$_{SMALL}$ (Turc et al., 2019). Finally, we described two usecases that leverage this embedding. Prototype tools for these applications can be obtained at `https://github.com/gorjanradevski/macchina/`.

## Acknowledgements

We acknowledge funding from the Flemish Government under the Onderzoeksprogramma Artificiële Intelligentie (AI) Vlaanderen programme. This work is supported by KU Leuven Internal Funds under the MACCHINA project.

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
