# OpenReview forum: "Self-supervised context-aware Covid-19 document exploration through atlas grounding"
_aclweb.org/ACL/2020/Workshop/NLP-COVID — NLP-COVID-2020 Abstractonly_

### Official Review · AnonReviewer2 · 2020-04-22
**Interesting anatomical exploration of COVID articles using BERT-based grounding**

**Rating:** 7
**Confidence:** 4

**Review:**

The paper proposes a self-supervised approach to ground sentences in COVID-19 articles to a 3D human atlas. This is the first study to use self-supervision to map sentences in medical articles to anatomical locations in the 3D atlas space. Unlike the prior works that utilized the atlas space, they formulated the grounding as a regression problem, thereby allowing a better idea about the semantic closeness of the sentences to the target body locations. The study has also proposed the potential use of this grounding approach for data visualization and document retrieval use cases.

Strengths:
-- The study employs a fully self-supervised technique for grounding sentences to a 3D human atlas. This is neat and has a lot of potential applications beyond COVID-19.
-- This work takes into account the prevalence of multiple organs in the same sentence while calculating the loss while training.

Major issues:
1. As discussed as one of the limitations that user study needs to be done to validate the query tools, as it is difficult to verify the actual performance of the document retrieval tool from the current version. Moreover, since the distance between the centroids of the point clouds is used to retrieve the articles relevant to a query, it is difficult to interpret whether this approach is useful as this does not take into account which of the sentences in the abstract are more close to the query. Some sentences in the article might just be providing background information and the corresponding organs might not be of interest to the user. To address this, it may be considered to have a way to identify the sentences of interest.

2. A brief discussion of interpreting the results mainly to indicate any potential relation between the impact of using the masked technique and the evaluation metrics used can be included.

3. Although the proposed method definitely obviates the expensive annotation part, it might be good to further compare the evaluations with a supervised model to validate that self-supervision performs better than a fully supervised approach. At the least, it would be good to compare the performance of these models with supervised models used on other datasets in the previous works for grounding medical text to 3D space.

Minor issues:
1. As already pointed out in the discussion that the models have only been trained on single sentences, it needs to be investigated whether a similar loss contribution strategy would work well when multiple sentences are included in the training process as organs appearing in different sentences might be very different from each other.
2. An evaluation of the methods to illustrate their performance on sentences having rare or infrequent organs may be reported.
3. How is the 8000 sentences mark determined for tackling the imbalance of different organs?

---

> ### Author Response · Authors · 2020-05-05
> **Reply to Reviewer2**
>
> Hello! Thank you for taking the time to read our paper and provide the feedback. Below, we respond to each one of the issues you pointed out.
>
> Major issues:
> 1) The most immediate consideration for improving the approach would be using more sophisticated ways to perform retrieval by using entire point clouds as opposed to using just their centroids, starting with the simple extension of incorporating the point cloud spreads in a way where more compact point clouds (therefore representing more specific documents) are given an advantage.
> Additionally, in order to mitigate the effect of organ-unrelated sentences to retrieval, we are currently working on projecting such sentences (sentences without organ occurrences) to locations outside of the human body.  We are expanding the current training set with sentences that do not mention any organ, and train new models that are more robust compared to the ones currently in the paper. For such sentences, the target used in the loss function is a set of points evenly spread across the outside surface of the convex hull of the body, and train the model such that it learns that any sentence that does not contain mentions of an organ should be mapped to a location outside of the human body. The loss function, using these anchor points just outside of the body in the same way as organ points, would encourage the mappings of such sentences to move towards the nearest location outside of the body. During inference, sentences projected to the outside of the human body could then be removed from document point clouds, effectively allowing us to isolate sentences of interest.
> 2) The use of the masking during training is essential for achieving high results on the sentences where the organ keywords do not occur i.e. where we rely on the context. During the training process, the use of masking certainly reduces the result on the non-masked evaluations (distances, ior’s etc.). However, without masking, the model would learn a 1-to-1 correspondence between the organ keywords and the locations in the atlas. Consequently, in the masked evaluations, where we check whether the model has learned to associate the context with its corresponding organs, the obtained result would be dramatically lower. In the updated version, we can include an experiment where the model is trained without masking and report the results for each of the metrics.
> 3) To the best of our knowledge, there haven’t been works that ground text in 3D space corresponding to the human body.  Therefore, we could not obtain any previously annotated data that would be adequate for our setup, and given the current situation, it was not feasible to create any kind of new annotation that would require medical expertise. Considering the global pandemic, our main goal was to find a cheap and scalable technique that could be deployed rapidly, so we did not consider comparisons with supervised methods.
>
> Minor issues:
> 1) Using longer sentences/more than one sentence can definitely be beneficial since it includes more context for the model to learn. However, using longer sentences potentially increases the risk of having too many organs occurring in a sentence which is not ideal for our self-supervised setup. Namely, if we expand our work in that direction (which certainly is interesting) we would aim for using some form of label propagation, instead of just inferring the label according to an organ occurrence within the sentence.
> 2) Computing a per-organ IOR/distance i.e. an IOR/distance macro-average for each of the organs is certainly feasible and we can include it in the updated version.
> 3) The threshold of 8000 was chosen as roughly the number of occurrences of the second, third, fourth and the fifth most occurring organ.

---

### Official Review · AnonReviewer3 · 2020-05-03
**interesting exploration with unclear justification**

**Rating:** 4
**Confidence:** 3

**Review:**

This paper describes methods for learning a mapping between sentence vectors (as produced by the [CLS] token in BERT) and 3D points in the human body. The idea is that sentences that are about some anatomical site can be mapped to that position in the body for visualization. It is never completely clear from the paper why this needs to be learned, since the training pre-processing already is finding anatomical site mentions and essentially normalizing them to a set of body parts with known coordinates. The only thing I can figure is that they are using their pipeline as a way to generate labeled training data, and they intend for this to be applied to sentences without explicit anatomical site mentions. This is maybe supported by the fact that their training does include 50% masking of anatomical sites. In that case, there is a missing element here, which is that the vast majority of sentences in a document are not "about" any organ at all, so mapping them into organ space is meaningless. So for this algorithm to be applicable, one would first need to pre-process sentences somehow for whether they should go onto the organ mapping stage at all. And in any case, their example application presumably is just mapping the sentences that did have anatomical site terms. Overall, I think the logic of the approach needs to be spelled out more clearly. As it is, it feels like an interesting trick that one can learn this mapping but I just don't see why it's necessary.

Other issues:
- This mapping will surely generate "point clouds" around organs (i.e. sentences mentioning lung will be spatially distributed near the lungs rather than all mapping directly to the centroid of the lung). This may give an expert in anatomy some sense that these minor differences are anatomically meaningful where I believe it would be largely noise, and if some finer-grained meaning were there it would be impossible to distinguish from noise.
- Results tables are never referred to in the text
- There should be some discussion of what you "want" if there are multiple organ mentions. I certainly don't want some "average" position, but also don't want it to just show up in 1 position, it should probably show up in all 3 positions. But that is not a function anymore so not really something you can ask a neural network to do.
- When masking anatomical sites in the training data, do you ever use the same sentence in both the masked and unmasked form? I think it would be preferable with this dataset to randomly choose one form before training, rather than randomly choosing to mask each time an instance is loaded (which could result in different masking behavior for the same instance across epochs) because otherwise it could be memorized.
- I don't know if self supervised is the right term here. It's more like a clever way of generating supervised instances.

---

> ### Author Response · Authors · 2020-05-07
> **Reply to Reviewer3**
>
> Hello! Thank you for taking the time to read our paper and provide the feedback.
>
> The main reason for learning a contextual mapping between sentences and parts of the human body is to go beyond the look-up table feature. Therefore, stochastic masking is applied during training. By stochastically masking the keyword references to the organ locations, the model is forced to learn something that is more meaningful than a 1-to-1 correspondence between the keyword and the organ location. This is further motivated by the fact that a sentence in a document may make an implicit reference to an organ, albeit never explicitly mentioning the organ by its name.
> When it comes to sentences without any medical relevance, it would indeed be possible to improve the tools via a pre-processing step that would ensure that the model operates more closely within its domain, for example by performing named entity recognition on input sentences and discarding the ones that do not contain any entity types from the medical domain.
>
> Other issues:
> - Our method and its use of softmin in the training objective accommodates the learning of meaningful spatial mappings around the organ from co-occurrences of different organ terms (e.g. bladder and kidney), and, by the virtue of the masking, to the co-occurrence of one organ term and contextual information typical for some other organ (e.g. bladder and nephron).
> - For the moment, the discussion regarding the results is done within the caption of the tables, but we can certainly update the text so that the results in the table are discussed in the main text.
> - One of the main issues with having an approach that does not use any annotated data is indeed the ambiguity when there are multiple mentions of an organ. However, our goal was to quickly prototype a solution that will help in the exploration of Covid-19 related documents, and not go through the cumbersome process of annotating medical documents which is extremely time consuming. Therefore we create a setup where we rely on the model to learn the more appropriate target during training.
> - Stochastically masking the keywords is essential to our approach, and defining the masking upfront will make the problem trivial for every sentence left unmasked. Namely, any sentence that does not have a masked keyword has a direct 1-to-1 correspondence between the organ name and the location in the human atlas. On the other hand, by stochastically masking the keywords, the network has to balance between the 1-to-1 correspondence, which is definitely desirable when there is an organ mention, and the contextual correspondence, which is essential in a real-life scenario.
> - We think that self-supervised is an appropriate term in a sense that the annotation comes with the occurrences of organ terms which come with the text itself, similarly to the way language modeling task is often referred to as self supervised as it relies on automatically available information found in the structure of the language itself.

---

### Official Review · AnonReviewer1 · 2020-05-13
**Novel task with under-developed motivation**

**Rating:** 4
**Confidence:** 4

**Review:**

This paper presents a BERT-based model for solving the novel task of grounding sentences to 3D locations in a representation of a human male body. The authors induce training labels by automatically extracting a list of organs and their synonymous wordforms using the SciSpacy UmlsEntityLinker and extracting sentences containing at least one mention of an organ on their curated list. The ground-truth voxel coordinates are assigned using organ locations from a 3D atlas, and several BERT models are fine-tuned using a distance-based regression loss. The paper presents a novel task, is clearly written, and thoroughly explains their experimental choices and settings.

Issues:

1. One of the key premises of the paper is that physical proximity in the human body reflects functional and semantic relatedness to a significant degree. This statement needs more justification, especially since it is such an important part of the motivation for this work. Looking at a model of a human torso, it seems that the centroid of the left lung may be closer to the centroid of the stomach than to the centroid of the trachea. Similarly, the centroid of the stomach seems closer to that of the spleen and pancreas than of the large intestine.

2. It's not clear what is gained by performing this as a regression task to directly predict 3D coordinates instead of as a classification task that can then be transformed to 3D coordinates using the atlas (the ClassCenter model). The motivation seems to be that in cases when the classification model gets the wrong organ, the predicted 3D location is significantly worse than in the regression model (i.e. NVD-O for ClassCenter is higher). However, the classification model identifies the correct organ with much significantly higher frequency (i.e. IOR is higher). The overall effect is that even in the masked setting, the average per-sentence voxel distance is much lower for ClassCenter (i.e. taking the linear combination of NVD and NVD-O w.r.t. IOR).

3. The use case needs further justification. In what situations is specifying a position in a 3D atlas preferable to submitting a textual (or UMLS-keyword) query? Since the training data only uses the centroids of organs, is there a meaningful difference between querying for the centroid of the stomach vs. somewhere else within the organ? Are the relative positions of sentences embedded within the same organ semantically meaningful, or mostly a product of noise in the regression?

4. Representing abstracts as a point cloud composed of individual sentences seems like it would cause issues for the information retrieval task presented in section 7.2. What happens to the majority class of sentences that don't contain a mention of an organ, either implicitly or explicitly? Does the frequency of sentences containing an organ mention accurately represent the relevance of the document?

---

> ### Author Response · Authors · 2020-05-24
> **Reply to Reviewer1**
>
> Hello! Thank you for taking the time to read our paper and provide the feedback. Below, we respond to the issues you pointed out.
>
> 1) We do not argue that the spatial proximity is always going imply a higher degree of relatedness between terms than the standard semantic similarity inferred from their distribution within the corpus (shared contexts ect.), but we nevertheless believe that it does reflect similarity between concepts to a significant degree. Additionally, in certain contexts, e.g. surgeries, spatial proximity in itself can be more relevant than semantic relatedness. And given the inherent ability to visualize texts within the human body that comes with the method, we believe it to be interesting to the research community.
>
> 2) Our motivation for framing the task as regression is that a sentence concerning a certain organ should be mapped preferably to that organ, and if not, at least to the general vicinity of it. We argue that even though the classification model achieves a lower overall distance than the regression based model, the higher and intractable error that it makes when predicting the wrong organ makes it less suitable for general use, and especially for retrieval tasks that rely on representing larger texts as combinations of their sentence mappings.
>
> 3) We proposed the use of a 3D point query as an alternative to textual queries for which many works already exist. The training objective takes into the account all voxels of a given organ, and not just organ centroids, and accommodates the functionality where clicking on the upper region of an organ retrieves sentences referring to it in the context of organs above it, while clicking onto its lower region retrieves sentences that also mention the organs below it.
>
> 4) We agree that the retrieval task would benefit from a pre-processing step that would discard medically irrelevant sentences (eg. without any named entities from the medical domain) and from a refined distance measure between clusters, but we believe that this is not essential to the general principle of the method.

---

### Decision · Program_Chairs · 2020-06-06

**Decision:**

Accept (Abstract only)

**Comment:**

Reviewers agreed there were a number of novel ideas here, though the motivation needs to be clarified in places. The authors mitigated the latter concern somewhat in their responses, but the case for the justification underlying their method could still be strengthened. In its present form, we think think the authors will benefit from presenting the work as an abstract at the meeting, and we hope that this meeting provides opportunity for them to receive additional feedback on the work.